# KNOWLEDGE CARD: FILLING LLMs' KNOWLEDGE GAPS WITH PLUG-IN SPECIALIZED LANGUAGE MODELS

**Shangbin Feng**[1]     **Weijia Shi**[1]     **Yuyang Bai**[2]
**Vidhisha Balachandran**[3]     **Tianxing He**[1]     **Yulia Tsvetkov**[1]
[1]University of Washington   [2]Xi'an Jiaotong University   [3]Carnegie Mellon University
shangbin@cs.washington.edu

## ABSTRACT

By design, large language models (LLMs) are static general-purpose models, expensive to retrain or update frequently. As they are increasingly adopted for knowledge-intensive tasks, it becomes evident that these design choices lead to failures to generate factual, relevant, and up-to-date knowledge. To this end, we propose KNOWLEDGE CARD, a modular framework to plug in new factual and relevant knowledge into general-purpose LLMs. We first introduce *knowledge cards*—specialized language models trained on corpora from specific domains and sources. Knowledge cards serve as parametric repositories that are selected at inference time to generate background knowledge for the base LLM. We then propose three content selectors to dynamically select and retain information in documents generated by knowledge cards, specifically controlling for *relevance*, *brevity*, and *factuality* of outputs. Finally, we propose two complementary integration approaches to augment the base LLM with the (relevant, factual) knowledge curated from the specialized LMs. Through extensive experiments, we demonstrate that KNOWLEDGE CARD achieves state-of-the-art performance on six benchmark datasets. Ultimately, KNOWLEDGE CARD framework enables dynamic synthesis and updates of knowledge from diverse domains. Its modularity will ensure that relevant knowledge can be continuously updated through the collective efforts of the research community. [1]

## 1 INTRODUCTION

Large language models (LLMs) have demonstrated an impressive ability to encode world knowledge in model parameters (Petroni et al., 2019; Roberts et al., 2020). However, they still face various challenges in knowledge-intensive tasks and contexts: they suffer from hallucination (Kryściński et al., 2020; Pagnoni et al., 2021; Ji et al., 2023), struggle to encode long-tail facts (Kandpal et al., 2023; Mallen et al., 2023), and could not be easily updated with new and emerging knowledge (De Cao et al., 2021; Hase et al., 2021). Existing works propose addressing these limitations through retrieval augmentation or generated knowledge prompting. *Retrieval-augmented LMs* (Guu et al., 2020; Borgeaud et al., 2022; Shi et al., 2023) employ retrieval systems to fetch relevant documents from a general and fixed retrieval corpus (e.g., Wikipedia or the Pile (Gao et al., 2020)), leveraging external knowledge from non-parametric sources to aid LLM generation. *Generated knowledge prompting* approaches (Shin et al., 2020; Liu et al., 2022a; Sun et al., 2022) prompt LLMs to incorporate and generate contextual documents to encourage knowledge-aware generation.

While the two lines of work have achieved some success, these existing systems struggle to reflect two key properties of knowledge. Knowledge is *modular* (Stuckenschmidt et al., 2009): it is an "archipelago" rather than a single "continent", encapsulating information that exists in diversified forms, domains, sources, perspectives, and more. The lack of knowledge modularity has made generalization to new domains and targeted updates of knowledge stored in LMs difficult. Knowledge is *collaborative* (Cayzer, 2004): LLMs should be able to represent and incorporate diverse and evolving knowledge, from multi-faceted sources and perspectives, while enabling collaborative contribution from various stakeholders. Community-driven knowledge could aggregate new knowledge from domain experts and enable the development of specialized LLMs, tailored to specific industries or applications. That being said, existing approaches and systems did not employ *modular* or *collaborative* knowledge sources that enable the plug-and-play updates and contributions from various stakeholders. While approaches such as retrieval augmentation could be extended for modularity,

---

[1]Resources are available at https://github.com/BunsenFeng/Knowledge_Card.

they are hardly compatible with the current landscape of model sharing (Wolf et al., 2019) and do not facilitate community-driven efforts to fill in LLMs' knowledge gaps.

To this end, we propose **KNOWLEDGE CARD**, a novel framework to empower general-purpose LLMs with modular and collaboratively-sourced knowledge through the integration of smaller, but specialized language models. As an increasing amount of powerful LLMs are released behind API calls, not directly accessible, and are prohibitively expensive to train or adapt, KNOWLEDGE CARD specifically focuses on augmenting black-box LLMs to enrich their knowledge capabilities. We first curate specialized LMs, *knowledge cards*, trained on corpora from diverse sources and domains to serve as modular knowledge repositories (§2.1). Compared to existing approaches, knowledge cards enable flexible and targeted information access, searching over domains, and employing private and personalized knowledge sources. These specialized LMs are later prompted to generate background information to support general-purpose LLMs. We then propose three levels of *knowledge selectors* to dynamically select and refine generated documents and control for topic relevance, document brevity, and knowledge factuality (§2.2). Finally, we propose *bottom-up* and *top-down*—two approaches to empower general-purpose LLMs by integrating outputs from specialized LMs (i.e.,plugging in knowledge cards into the LLM) (§2.3). Specifically, the *bottom-up* approach starts by prompting all knowledge cards to generate multiple documents, then performs selection with the three knowledge selectors, while concatenating the final knowledge paragraph with the query for LLM generation. While the bottom-up approach uniquely enables multi-domain knowledge synthesis, it also presents the risk of presenting irrelevant information to LLM in contexts where external information is not needed. This motivates us to propose the *top-down* approach, where the general-purpose LLM itself decides whether external knowledge is necessary for the given query, then relevant knowledge cards are selectively activated for knowledge integration; this process is repeated until the general-purpose LLM has enough confidence to generate a response.

Extensive experiments demonstrate that KNOWLEDGE CARD outperforms vanilla LLMs, retrieval-augmented LMs, and generated prompting approaches on three tasks across six datasets. For *general-purpose knowledge QA*, KNOWLEDGE CARD improves Codex performance by 6.6% on MMLU and even outperforms the 3-times larger Flan-PaLM. For *misinformation analysis* that tests multi-domain knowledge integration, KNOWLEDGE CARD outperforms all baseline approaches by at least 15.8% and 10.0% balanced accuracy scores on two- and four-way classification settings. In the third task, to evaluate the ability to update the knowledge of general-purpose LLMs, we curate MIDTERMQA, a QA dataset focusing on the 2022 U.S. midterm elections while the knowledge cutoff of LLMs is generally 2021 or earlier. Experiments demonstrate that KNOWLEDGE CARD outperforms all baselines by at least 55.6% on exact match scores, showcasing the ability for temporal knowledge update while only adding one knowledge card trained on midterm election news with 100x fewer parameters than the general-purpose LLM. Our findings demonstrate the potential of filling in the knowledge gaps of general-purpose LLMs by integrating modular and collaborative knowledge from small, independently trained, and specialized LMs. We envision KNOWLEDGE CARD as an initiative to encourage LM developers to collaborate in expanding the knowledge of large language models while reducing the carbon footprint from retraining gigantic LMs from scratch.

## 2 METHODOLOGY

We introduce KNOWLEDGE CARD, a novel framework to empower general-purpose LLMs with modular and collaborative knowledge (Figure 1). We train various *knowledge cards*, LMs trained on specialized knowledge corpora from diversified domains and sources (§2.1). We then use them to produce background knowledge for the general-purpose LLMs, while employing three *knowledge selectors* to ensure quality in knowledge synthesis (§2.2). Finally, we propose *bottom-up* and *top-down*, two approaches to condition the LLM on the content sourced from knowledge cards and post-processed using the knowledge selectors (§2.3).

### 2.1 KNOWLEDGE CARDS

While existing approaches rely on one fixed source of knowledge to improve LLMs (one retrieval corpus (Guu et al., 2020; Borgeaud et al., 2022; Shi et al., 2023), one knowledge graph (Wang et al., 2021; Zhang et al., 2021; Feng et al., 2023c), or one pretrained LLM itself (Shin et al., 2020; Liu et al., 2022a; Sun et al., 2022)), we hypothesize that since knowledge is modular, general-purpose LLMs should be augmented with modular plug-and-play knowledge repositories that allow users to collaboratively add, remove, edit, or update information. In addition, different communities might

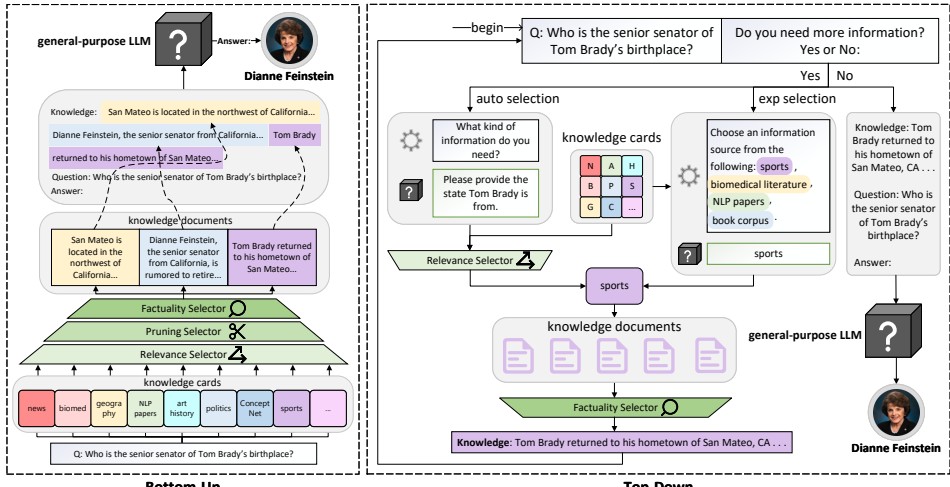

Figure 1: Overview of KNOWLEDGE CARD. We train knowledge cards on various knowledge domains and employ three knowledge selectors for quality control. We propose *bottom-up* and *top-down* to integrate general-purpose LLMs with modular and specialized LMs for multi-domain knowledge synthesis (*bottom-up*) and proactively seeking external knowledge (*top-down*).

have different definitions and requirements for knowledge. Wikipedia factoids, biomedical literature, mathematical formulae, and commonsense knowledge graphs are all valuable knowledge components in various contexts, thus LLMs should be able to represent and incorporate knowledge contributed by stakeholders across multi-faceted domains and industries.

To this end, we propose to curate *knowledge cards*, specialized LMs that are much smaller than black-box LLMs, trained on diversified knowledge corpora from a wide range of domains and sources. Concretely, we obtain $n$ knowledge cards $\mathcal{C} = \{c_1, c_2, \cdots, c_n\}$, each starting from an existing LM checkpoint and further trained on a specific knowledge corpora $\mathcal{D}_i$ with the causal language modeling objective. Given a query to the LLM, these knowledge cards are selectively activated and used with prompted generation. Formally, given query $q$, specialized LM $c$ defines a mapping $c(q) : q \rightarrow d_q$ where $q$ is used as prompt to generate a continuation as the knowledge document $d_q$, which are later prepended into the context of general-purpose LLMs through various mechanisms (§2.3).

In this way, the modularity of knowledge is demonstrated through the effortless addition, removal, or selective activation of various knowledge cards during the LLM generation process. Similarly, the collaborative nature of knowledge is reflected by enabling individuals to contribute trained knowledge cards on their desired knowledge source to KNOWLEDGE CARD, expanding the knowledge of general-purpose LLMs through community-driven efforts.

## 2.2 KNOWLEDGE SELECTORS

While it is possible to directly adopt $d_q$ as relevant knowledge, we identify three key challenges in the successful integration of knowledge cards and general-purpose LLMs: relevance, brevity, and factuality. We design three respective selectors to control for such factors.

**Relevance Selector** While we expect knowledge cards to generate background information that is relevant and helpful to the query $q$, LMs sometimes deviate from the query (Holtzman et al., 2019). Furthermore, only a handful of knowledge cards would be relevant for a given query. To this end, we propose to select and retain knowledge documents based on relevance. Concretely, given a set of $m$ generated documents $\{d_1, \cdots, d_m\}$ and the query $q$, we aim to retain the top-$k$ relevant documents and discard irrelevant information. We adopt a separate encoder-based LM $\text{enc}(\cdot)$ that maps a token sequence to a feature vector and cosine similarity $\text{sim}(\cdot, \cdot)$ to measure relevance. Formally, we retain $d_i$ if $i \in \text{top-k}_j(\text{sim}(\text{enc}(d_j), \text{enc}(q)))$ where top-k is the top-$k$ $\text{argmax}$ operation.

**Pruning Selector** Existing works mostly integrate one piece of external knowledge into LLMs (Sun et al., 2022; Shi et al., 2023), while tasks requiring integration of multiple domains of information,

such as misinformation detection (Karimi et al., 2018) and multi-hop QA (Nishida et al., 2019), are not well supported by existing paradigms. To effectively incorporate generated documents from multiple LMs while fitting into the LLM context length limit, we propose to prune knowledge documents. Formally, given $m$ documents $\{\boldsymbol{d}_1, \cdots, \boldsymbol{d}_m\}$, we adopt a pruning model $\mathrm{prune}(\cdot)$, operationalized most simply as a summarization system (Zhang et al., 2020; Liu et al., 2022b), to obtain the condensed versions separately $\{\tilde{\boldsymbol{d}}_1, \cdots, \tilde{\boldsymbol{d}}_m\}$. This pruning method allows for the integration into the main LLM of information from multiple domains while preserving space for in-context learning.

**Factuality Selector** Language models are prone to hallucination (Ji et al., 2023) and the knowledge cards are no exception. Given a set of $m$ pruned knowledge documents $\{\tilde{\boldsymbol{d}}_1, \cdots, \tilde{\boldsymbol{d}}_m\}$, their original versions $\{\boldsymbol{d}_1, \cdots, \boldsymbol{d}_m\}$, and the query $\boldsymbol{q}$, we filter out the non-factual knowledge and retain $\ell$ documents. Specifically, we evaluate the factuality of knowledge documents with two measures.

We first evaluate *summarization factuality*, ensuring that the pruned version $\tilde{\boldsymbol{d}}_i$ factually captures the important points in the original $\boldsymbol{d}_i$. Concretely, we adopt factuality evaluation models (Kryściński et al., 2020; Feng et al., 2023a) as a scoring function $\mathrm{sum\text{-}fact}(\cdot, \cdot)$, where each knowledge document $\boldsymbol{d}$ is assigned a summarization factuality score $s_{\boldsymbol{d}}^{sum} = \mathrm{sum\text{-}fact}(\tilde{\boldsymbol{d}} \mid \boldsymbol{d}) \in [0, 1]$.

We then propose to evaluate whether the generated knowledge document is well-supported by real-world knowledge through *retrieval-augmented fact checking*. Specifically, given a knowledge document $\boldsymbol{d}$, we retrieve $k$ documents from a retrieval corpus $\boldsymbol{t}_1, \ldots, \boldsymbol{t}_k$, then employ a fact-checking model (Schuster et al., 2021) as a scoring function $\mathrm{fact\text{-}check}(\cdot, \cdot)$. We then assign a fact-checked factuality score to each $\boldsymbol{d}$ based on the retrieved document that *most* supports $\boldsymbol{d}$, formally $s_{\boldsymbol{d}}^{fact} = \max_{1 \le i \le k} \mathrm{fact\text{-}check}(\boldsymbol{d} \mid \boldsymbol{t}_i) \in [0, 1]$. We then average the summarization factuality score and the fact-checking score for each document to obtain $s_{\boldsymbol{d}}$.

While it is straightforward to greedily select $\ell$ knowledge documents with the highest $s_{\boldsymbol{d}}$ scores, new and more recent knowledge might not be well-supported by existing fact-checking tools. As a result, we propose *top-$k$ factuality sampling* to allow for flexibility while remaining stringent towards knowledge documents that are clearly wrong. Formally, we first obtain $\mathcal{D}^k$ as the set of knowledge documents with the top-$k$ factuality scores where $k > \ell$ is a hyperparameter. We then define a sampling probability distribution over all $m$ knowledge documents:

$$p(\tilde{\boldsymbol{d}}_i \mid \boldsymbol{q}) = \begin{cases} \exp(s_{\boldsymbol{d}_i}) / \sum_{\boldsymbol{d}_j \in \mathcal{D}^k} \exp(s_{\boldsymbol{d}_j}), & \text{if } \tilde{\boldsymbol{d}}_i \in \mathcal{D}^k. \\ 0, & \text{if } \tilde{\boldsymbol{d}}_i \notin \mathcal{D}^k. \end{cases}$$

We sample $\ell$ knowledge documents from $\{\tilde{\boldsymbol{d}}_1, \cdots, \tilde{\boldsymbol{d}}_m\}$ with probabilities $\{p(\tilde{\boldsymbol{d}}_1 \mid \boldsymbol{q}), \cdots, p(\tilde{\boldsymbol{d}_m} \mid \boldsymbol{q})\}$. In this way, knowledge documents with very low factuality scores are strictly removed while flexibility is built in through sampling from the knowledge with factuality scores near the top.

## 2.3 Knowledge Integration

After defining the modular components in Knowledge Card (a general-purpose LLM, knowledge cards, and knowledge selectors), we propose two approaches, *bottom-up* and *top-down*, to integrate the general-purpose LLM with external knowledge sources, which are selected outputs of knowledge cards. Specifically, *bottom-up* activates all available knowledge cards at once and employs the three knowledge selectors to control for knowledge quality. *Bottom-up* enables multi-domain knowledge synthesis across all available sources, but these might occasionally introduce irrelevant information which may adversely impact LLM inference. We additionally propose a *top-down* approach, in which the LLM proactively seeks external information from selected knowledge cards. *top-down* is advantageous in tasks and domains where external knowledge is not always necessary.

**Bottom-Up Approach** *Bottom-up* starts by prompting available knowledge cards, then progressively goes through the three knowledge selectors, and these outputs are incorporated into the LLM via the prompt context. Formally, given $n$ knowledge cards $\mathcal{C} = \{\boldsymbol{c}_1, \cdots, \boldsymbol{c}_n\}$ and the query $\boldsymbol{q}$, we generate $n_1$ documents with each knowledge card through temperature sampling (Holtzman et al., 2019) to obtain $\{\boldsymbol{d}_1, \cdots, \boldsymbol{d}_{n \times n_1}\}$. We first apply the relevance selector to retain $n_2$ most relevant documents $\{\boldsymbol{d}_1, \cdots, \boldsymbol{d}_{n_2}\}$, then conduct knowledge pruning through the pruning selector $\{\tilde{\boldsymbol{d}}_1, \cdots, \tilde{\boldsymbol{d}}_{n_2}\}$, and finally leverage the factuality selector to obtain $n_3$ high-quality knowledge documents $\{\tilde{\boldsymbol{d}}_1, \cdots, \tilde{\boldsymbol{d}}_{n_3}\}$.

The final prompt for the LLM is a concatenation of knowledge documents and the query, formally ["*Knowledge:* " $\| \tilde{d}_1 \| \tilde{d}_2 \| \cdots \| \tilde{d}_{n_3} \| q$] where $\|$ denotes concatenation. We expect the bottom-up approach to be strong in multi-domain knowledge synthesis since multiple knowledge cards could be activated at once to provide background knowledge from diverse perspectives. In addition, hyperparameters $n_1$, $n_2$, and $n_3$ enable fine-grained control over the knowledge synthesis process.

**Top-Down Approach** In *bottom-up*, we assume that every query would benefit from external knowledge generated by knowledge cards. However, this could introduce unnecessary information in the LLM's prompt context (Zhao et al., 2023). Following Kadavath et al. (2022), who showed that LLMs possess preliminary abilities to identify their inherent knowledge limitations, we propose the *top-down* approach, putting the LLM in charge to iteratively identify whether external knowledge is needed and selectively activate relevant knowledge cards through various strategies.

Concretely, for the $n$ knowledge cards $\mathcal{C} = \{c_1, \cdots, c_n\}$, we also ask the knowledge card contributors to submit a textual description of LMs $\mathcal{S} = \{s_1, \cdots, s_n\}$ such as "*biomedical literature*", "*college calculus*", or "*commonsense knowledge graph*". We first ask the LLM a yes/no question to determine whether external knowledge is needed for the given query $q$, specifically "*Do you need more information? (Yes or No)*". We encourage better-calibrated answers to the yes/no question through in-context learning (Wei et al.; Press et al., 2022): specifically, we introduce a set of in-context learning examples that encompass two distinct categories of questions posed to the LLM. The first category consists of questions that the LLM is capable of answering accurately without the need for any extra information. For these questions, the response to the query "Do you need more information (Yes or No)?" is "No." The second category comprises questions that the LLM cannot answer correctly without the provision of additional information. In this case, the corresponding output label for the query is "Yes." In this way, we prompt the LLM to learn to request external knowledge through in-context learning; we analyze the effectiveness of this approach in Section 5. If the LLM answers "*No*", we directly prompt the LLM to generate based on the query, without resorting to knowledge cards. If the LLM requests external knowledge by answering "*Yes*", we employ two strategies (Algorithm 2) to select a relevant knowledge card and generate background knowledge.

- **Automatic Selection** (AUTO) We further prompt the LLM with "*What kind of information do you need?*" and select one knowledge card based on its response $r_q$. Concretely, we identify which LM description $\{s_1, \cdots, s_n\}$ is most relevant to $r_q$ with the relevance selector (§2.2) and activate the corresponding LM to generate multiple knowledge documents, then select one with the highest factuality score based on the factuality selector (§2.2) to obtain $d$.

- **Explicit Selection** (EXP) Alternatively, we ask the LLM to directly select one knowledge card by prompting with "*Choose an information source from the following: $s_1, \ldots, s_n$*". If the LLM responds with $s_i$, we activate the corresponding knowledge card $c_i$ to generate multiple knowledge documents and select one with the factuality selector (§2.2) to obtain $d$.

Upon obtaining the document, we append "*Knowledge: $d$*" to the LLM context. We then iteratively ask "*Do you need more information? (Yes or No)*" again, repeat the above process, until the LLM answers "*No*" and generates a knowledge-informed response. We expect *top-down* to perform better when external knowledge is not always necessary. In this way, the top-down approach enables LLMs to take charge in identifying their inherent knowledge limitations and seeking help from external knowledge cards proactively. We provide prompt examples in Tables 10 and 11 in the Appendix.

## 3 EXPERIMENT SETTINGS

**Implementation** For *knowledge cards*, we use OPT-1.3B (Zhang et al., 2022) as the starting point and separately train 25 specialized LMs on a wide range of knowledge sources and domains, including corpora in the Pile (Gao et al., 2020), branch-train-merge (Li et al., 2022), knowledge graphs (Speer et al., 2017; West et al., 2022; Vrandečić & Krötzsch, 2014; Pellissier Tanon et al., 2020; Feng et al., 2021; Zhang et al., 2021), news and social media (Liu et al., 2022c; Feng et al., 2023b), and more. (Appendix E) We use MPNet (Song et al., 2020) as the encoder in the *relevance selector*, Pegasus (Zhang et al., 2020) as the summarization model in the *pruning selector*, the WikiSearch API as the retrieval system in the *factuality selector*, and FactKB (Feng et al., 2023a) and VitaminC (Schuster et al., 2021) as the summarization and fact-checking factuality scoring functions. We use Codex (CODE-DAVINCI-002) (Chen et al., 2021) as the default, general-purpose, black-box LLM.

| Type | Model | Human. | Social | STEM | Other | All |
|------|-------|--------|--------|------|-------|-----|
| **Vanilla LM** | CODEX | 74.2 | 76.9 | 57.8 | 70.1 | 68.3 |
| | PALM | 77.0 | 81.0 | 55.6 | 69.6 | 69.3 |
| | FLAN-PALM | - | - | - | - | 72.2 |
| **Retrieval** | ATLAS | 46.1 | 54.6 | 38.8 | 52.8 | 47.9 |
| | REPLUG | 76.0 | 79.7 | 58.8 | 72.1 | 71.4 |
| | REPLUG LSR | 76.5 | 79.9 | 58.9 | 73.2 | 71.8 |
| **Generate** | GKP | 73.3 | 74.5 | 59.5 | 71.4 | 70.0 |
| | RECITATION | 76.9 | 78.1 | 59.0 | 74.0 | 71.9 |
| **KNOWLEDGE CARD** | BOTTOM-UP | 77.2 | 76.7 | 57.9 | 72.2 | 70.7 |
| | TOP-DOWN AUTO | 77.7 | 78.9 | 59.2 | 73.0 | 72.0 |
| | TOP-DOWN EXP | **78.6** | **80.9** | **59.6** | **74.3** | **72.8** |

Table 1: Model performance on the MMLU Benchmark. KNOWLEDGE CARD improves Codex by at least 3.5% while *top-down* outperforms all baselines.

| Type | Model | Two-Way BAcc | Two-Way MaF | Four-Way BAcc | Four-Way MaF |
|------|-------|------|-----|------|-----|
| **Vanilla LM** | CODEX | 65.6 | 51.0 | 52.8 | 44.0 |
| **Retrieval** | REPLUG | 78.8 | 67.8 | 55.8 | 53.0 |
| | REPLUG LSR | 78.8 | 68.5 | 57.5 | 54.4 |
| **Generate** | GKP | 73.5 | 60.3 | 61.1 | 46.3 |
| | RECITATION | 65.0 | 47.7 | 64.2 | 48.6 |
| | GRTR | 66.1 | 49.1 | 51.6 | 36.9 |
| **KNOWLEDGE CARD** | BOTTOM-UP | 89.8 | **87.3** | **70.6** | **67.3** |
| | TOP-DOWN AUTO | 86.4 | 78.7 | 63.0 | 60.2 |
| | TOP-DOWN EXP | **91.3** | 86.0 | 69.4 | 65.5 |

Table 2: Performance on misinformation detection. BAcc and MaF are balanced accuracy and macro F1. *bottom-up* performs best due to multi-domain knowledge integration.

| Type | Model | Open-Book EM | Open-Book F1 | Multiple-Choice 2-way | Multiple-Choice 4-way |
|------|-------|------|-----|------|------|
| **Vanilla LM** | CODEX | 55.1 | 57.9 | 90.9 | 60.8 |
| **Retrieval** | REPLUG | 44.8 | - | 85.7 | 62.8 |
| | REPLUG LSR | 37.2 | - | 86.9 | 65.3 |
| | SI ET AL. | 52.1 | 54.5 | 84.7 | 61.4 |
| **Generate** | GKP | 45.0 | 46.9 | 89.1 | 53.5 |
| | RECITATION | 44.4 | 46.4 | 89.3 | 52.3 |
| | GRTR | 55.6 | 58.4 | 77.4 | 59.0 |
| **KNOWLEDGE CARD** | BOTTOM-UP | 83.6 | 85.6 | 81.6 | 64.5 |
| | TOP-DOWN AUTO | 87.5 | 89.3 | 89.5 | 63.0 |
| | TOP-DOWN EXP | 75.3 | 75.7 | **91.9** | **67.6** |

Table 3: Performance on MidtermQA. KNOWLEDGE CARD successfully updates the knowledge of Codex by adding a single knowledge card.

**Tasks and Datasets** 1) For *general-purpose QA*, we adopt MMLU (Hendrycks et al., 2020), a multiple-choice QA dataset covering 57 tasks in humanities, STEM, social sciences, and others. Following previous works (Si et al., 2022; Shi et al., 2023), we adopt a 5-shot in-context learning setting. 2) To evaluate *multi-domain knowledge synthesis*, we adopt misinformation detection, since news articles often encompass facts and opinions at the intersection of different domains and perspectives. We leverage the widely adopted LUN misinformation detection dataset (Rashkin et al., 2017) with both 2-way and 4-way classification settings. All models are evaluated based on 16-shot in-context learning. 3) To evaluate *temporal knowledge update*, we curate MIDTERMQA, a QA benchmark focusing on the 2022 U.S. midterm elections since the knowledge cutoff of black-box LLMs is often 2021 or earlier. MIDTERMQA presents three evaluation datasets and settings: open-book, 2-way, and 4-way multiple choice. 5-shot in-context learning is adopted to evaluate KNOWLEDGE CARD and baselines. We did not consider existing temporal QA datasets (Jang et al., 2021; Dhingra et al., 2022; Kasai et al., 2022) since they do not focus on any specific event or knowledge domain.

**Baselines** We compare KNOWLEDGE CARD with a wide range of baseline methods in three categories. 1) vanilla black-box LLMs: Codex (Chen et al., 2021), PaLM (Chowdhery et al., 2022), and Flan-PaLM (Chung et al., 2022); 2) generated knowledge prompting approaches: GKP (Liu et al., 2022a), recitation (Sun et al., 2022), GRTR (Yu et al., 2022) (Note that we apply these methods to the same LLM Codex (Chen et al., 2021) for a fair comparison); 3) retrieval-augmented language models: Atlas (Izacard et al., 2022), Si et al. (2022), RePlug, and RePlug LSR (Shi et al., 2023).

## 4 RESULTS

**MMLU** For general-purpose knowledge QA, we use the MMLU benchmark (Hendrycks et al., 2020). As shown in Table 1, all three configurations of KNOWLEDGE CARD significantly improve vanilla Codex. Among them, the top-down approach with explicit selection performs best, improving Codex by 6.6% overall accuracy. Concurrently, top-down approaches surpass all baselines, including Flan-PaLM with a few hundred billion more parameters. These results suggest that we present an effective approach for making general-purpose LLMs better in knowledge-intensive contexts. In addition, *top-down* generally outperforms *bottom-up* likely because MMLU contains math-related questions that do not necessitate external knowledge. This observation suggests that *top-down* approaches are better at tasks where external knowledge is not always necessary.

**Misinformation Detection** To examine whether KNOWLEDGE CARD successfully integrates multi-faceted knowledge from diversified sources, we adopt the LUN misinformation dataset (Rashkin et al., 2017) with two- and four-way classification settings. Table 2 demonstrates that KNOWLEDGE CARD significantly improves Codex by at least 31.7% and 19.4% in balanced accuracy scores for both settings. In addition, *bottom-up* outperforms both variants of *top-down*, thanks to its methodology to jointly activate knowledge cards from various domains and enable multi-domain knowledge synthesis.

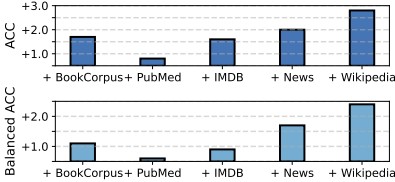

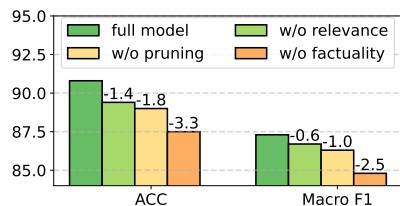

Figure 2: Performance on misinformation detection when each knowledge card is separately added. KNOWLEDGE CARD enables modular patching of LLMs while in-domain knowledge cards help the most.

Figure 3: Ablation study of the three knowledge selectors on misinformation detection. While the three selectors all contribute to model performance, the *factuality* selector is most crucial.

**MidtermQA**   To examine whether KNOWLEDGE CARD could update the parametric knowledge of LLMs, we train an additional knowledge card on news articles regarding the 2022 U.S. midterm elections and plug it into KNOWLEDGE CARD. We present model performance on MidtermQA in Table 3, which demonstrates that KNOWLEDGE CARD substantially outperforms all baselines in the open-book setting by as much as 57.3% in exact match scores (EM). This indicates that one knowledge card with 1.3B parameters successfully updates the parametric knowledge of the 175B Codex through KNOWLEDGE CARD. In addition, *top-down* outperforms *bottom-up*, indicating that the selective activation of knowledge cards is better when there is a specific knowledge card tied to the task domain. KNOWLEDGE CARD also outperforms SI ET AL. (Codex + Contriever) that uses the same midterm election news as retrieval corpora. In addition, generated knowledge prompting approaches (GKP, recitation, GRTR) underperform vanilla Codex, showing that probing LLMs for explicit knowledge is counterproductive when internal LLM knowledge is outdated or wrong.

## 5 ANALYSIS

**Patching LLM Knowledge**   When general-purpose LLMs struggle at tasks due to knowledge limitations, KNOWLEDGE CARD could serve as an efficient approach to patch LLM weaknesses by adding specialized language models. To this end, we evaluate the change in performance when five knowledge cards are separately added to augment Codex with the top-down approach. Results in Figure 2 demonstrate that patching the LLM with all five LMs leads to various levels of performance gains on misinformation detection, while the most in-domain LMs (Wikipedia and news) lead to greater improvements. This suggests that when LLMs perform poorly on knowledge-intensive tasks, an additional knowledge card trained on in-domain corpora could help with KNOWLEDGE CARD.

| Model | Two-Way | | Four-Way | |
|---|---|---|---|---|
| | BAcc | MaF | BAcc | MaF |
| REPLUG | 78.8 | 67.8 | 55.8 | 53.0 |
| REPLUG LSR | 78.8 | 68.5 | 57.5 | 54.4 |
| BOTTOM-UP | **90.0** | **87.0** | **65.3** | **63.3** |
| TOP-DOWN AUTO | 80.7 | 70.9 | 60.1 | 56.8 |
| TOP-DOWN EXP | 80.6 | 70.0 | 59.7 | 56.5 |

Table 4: KNOWLEDGE CARD outperforms retrieval LM REPLUG in the Wikipedia-only setting, suggesting that modular LMs present a better knowledge repository than retrieval.

**Knowledge Selector Study**   In Section 2.2, we propose three levels of knowledge selectors to control for various factors and ensure knowledge quality. We conduct ablation studies to remove each knowledge selector in the bottom-up approach and re-evaluate on misinformation detection. Figure 3 demonstrates that while all three knowledge selectors are helpful, the factuality selector contributes most to model performance and thus plays a crucial role in ensuring the quality of generated knowledge documents.

**Retrieval vs. Specialized LMs**   In order to assess the effectiveness of modular specialized LMs as compared to non-parametric sources like retrieval, we exclusively use the Wikipedia LM in KNOWLEDGE CARD and compare with the state-of-the-art retrieval LM REPLUG that also uses Wikipedia as the retrieval knowledge source. Table 4 demonstrates that KNOWLEDGE CARD outperforms REPLUG on both settings of misinformation detection, suggesting that knowledge cards present a better knowledge repository. Note that KNOWLEDGE CARD is also *compatible* with multiple knowledge formats (e.g. retrieval and search engine) while they could be complementary (Appendix A).

**Knowledge Stream Analysis**   In *bottom-up*, three hyperparameters (§2.3) govern the "knowledge stream" from knowledge cards to the general-purpose LLMs. Specifically, $n_1$ controls how many

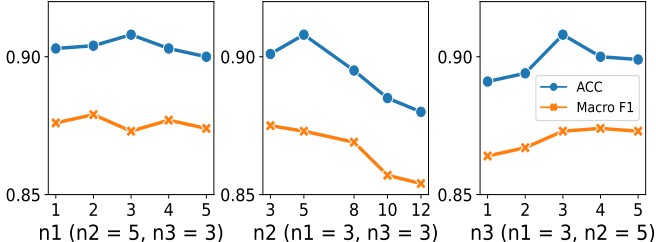

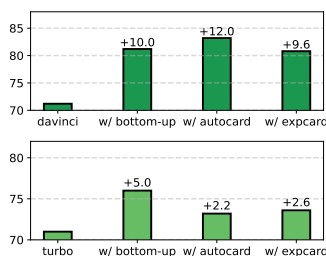

Figure 4: Investigating the impact of $n_1$, $n_2$, and $n_3$, which govern the knowledge stream from modular knowledge cards to general-purpose LLMs. These hyperparameters enable fine-grained control over the knowledge synthesis process.

Figure 5: KNOWLEDGE CARD is compatible with other LLMs, specifically TEXT-DAVINCI-003 and GPT-3.5-TURBO.

documents each LM generates, $n_2$ controls how many are retained after the three knowledge selectors, and $n_3$ controls how many are put into the context of LLMs. We investigate these control measures and report performance in Figure 4. It is illustrated that: 1) $n_1$ has a marginal impact, suggesting that knowledge cards generate largely homogeneous knowledge even with temperature sampling (Caccia et al., 2018); 2) larger $n_2$ leads to performance drops, suggesting that the three knowledge selectors ensure knowledge quality; 3) $n3 = 1$, where only one knowledge document is adopted at a time (as in previous works (Sun et al., 2022; Shi et al., 2023)) is worse than larger values, showing the advantage of multi-domain knowledge synthesis uniquely enabled by KNOWLEDGE CARD.

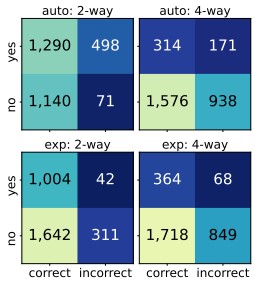

Figure 6: Confusion matrices of yes/no and correctness in *top-down*, enabling fine-grained error analysis.

**LLM Compatibility** While we follow previous works (Sun et al., 2022; Shi et al., 2023) and adopt Codex as the default black-box LLM, KNOWLEDGE CARD is compatible with different models. We additionally evaluate KNOWLEDGE CARD with two other LLMs, TEXT-DAVINCI-003 and GPT-3.5-TURBO, and present results in Figure 5. Both *bottom-up* and *top-down* consistently improve different LLMs across various datasets and evaluation metrics.

**Yes/No in *Top-Down*** In *top-down* (§2.3), we begin by asking LLMs if they might need external knowledge for the given query and adopt in-context examples to encourage well-calibrated answers. We illustrate LLM responses along with the correctness of their answer in Figure 6. The vast majority of queries are mapped to the "yes, correct" and "no, correct" categories, suggesting that LLMs have preliminary abilities to "know what they know" and seek external information if necessary. However, this ability is far from perfect, evident in the non-negligible category of "no, incorrect", suggesting that prompting LLMs to acknowledge knowledge limitations requires further research (Kadavath et al., 2022; Zhao et al., 2023), while new approaches to abstain could be easily integrated into KNOWLEDGE CARD. In addition, the "yes, incorrect" categories suggest that specialized LMs occasionally fail to provide enough information. These confusion matrices provide fine-grained error analysis and guidance as to whether the general-purpose LLM, the yes/no question, or knowledge cards require further improvements.

| Race | Codex | KNOWLEDGE CARD |
|---|---|---|
| AL, senate | Doug Jones ✗ | Katie Britt ✓ |
| PA, senate | Bob Casey ✗ | John Fetterman ✓ |
| CA, 3rd | Mike Thompson ✗ | Kevin Kiley ✓ |
| IN, 2nd | Jackie Walorski ✗ | Jim Banks ✗ |
| NV, governor | Steve Sisolak ✗ | Joe Lombardo ✓ |

Table 5: While vanilla Codex falsely claims that these incumbents won again in the 2022 elections, KNOWLEDGE CARD successfully updates the knowledge of black-box LLMs.

**Qualitative Analysis** We curated MIDTERMQA to evaluate whether KNOWLEDGE CARD enables efficient knowledge update. We examine the 88 races where the incumbent was not re-elected: Codex answered 1 out of the 88 questions correctly, while *bottom-up* and *top-down* with automatic and explicit selection answered 63, 77, and 42 correctly. Table 5 shows that Codex states the incumbents would win again in 2022, while KNOWLEDGE CARD successfully updates LLMs with 100x more parameters.

## 6 RELATED WORK

**Retrieval-Augmented Language Models** Augmenting language models with retrieval has advanced the state-of-the-art in open-domain QA (Guu et al., 2020; Izacard et al., 2022; Lewis et al., 2020; Hu et al., 2022), text classification (Zhao et al., 2023), and language modeling (Hu et al., 2022; Borgeaud et al., 2022; Min et al., 2023). The retrieval system could be integrated into encoder-decoder (Izacard et al., 2022) and decoder-only models (Borgeaud et al., 2022; Shi et al., 2022; Rubin et al., 2022), or leveraged to interpolate the next token probability distributions (Khandelwal et al., 2019; Zhong et al., 2022). Recent advances incorporated frozen (Mallen et al., 2023; Si et al., 2022; Khattab et al., 2022) and trainable retrievers (Shi et al., 2023) as well as search engines (Press et al., 2022) to augment LLMs. Compared to retrieval models and search engines, KNOWLEDGE CARD enables flexible information seeking, searching over knowledge domains, and employing private knowledge sources. In addition, these works often leverage only *one* retrieval corpora and assume that it's "omniscient" while suffering from various issues such as domain coverage and knowledge update. In contrast, we propose to reflect the modularity and community-driven nature of knowledge by integrating plug-and-play knowledge cards with general-purpose LLMs.

**Generated Knowledge Prompting** LMs acquire knowledge through training on gargantuan textual corpora (Petroni et al., 2019; Dhingra et al., 2022; He et al., 2021). Generated knowledge prompting (Liu et al., 2022a) is one of the early approaches to tap into the parametric knowledge of LLMs by prompting them to generate background information and re-using it for QA. Related works also propose to use LM parametric knowledge for retrieval (Tay et al., 2022), answer commonsense questions with self-talk (Shwartz et al., 2020), generate queries (Wang et al., 2022; Zhuang et al., 2022) or token sequences (Bevilacqua et al., 2022) for document augmentation. In addition, recitation-augmented language models (Sun et al., 2022) propose to augment QA examples with diversified knowledge recitations, while (Yu et al., 2022) shows that generated knowledge is, under certain circumstances, better than retrieval. However, this line of work assumes that the encoded knowledge in LLM parameters is all we need, while LLM knowledge suffers from hallucination (Ji et al., 2023), struggles to encode long-tail facts (Mallen et al., 2023), and can not be efficiently updated (De Cao et al., 2021). While recent works propose to edit LLM knowledge (Meng et al., 2022; Hernandez et al., 2023), they are hardly compatible with black-box LLMs. In addition, parametric knowledge in LLMs is far from modular and collaborative, while LMs should be able to incorporate knowledge contributed by all stakeholders in LLM research and applications. To this end, we propose KNOWLEDGE CARD as a community-driven initiative to empower general-purpose LLMs with modular and collaborative knowledge through the sharing and re-using of knowledge cards.

**Modular LMs** Mixture-of-Experts (MoE) (Masoudnia & Ebrahimpour, 2014) aims to activate one expert based on the input instance, which has been adopted in language model research (Gururangan et al., 2022; Roller et al., 2021; Lewis et al., 2021; Kudugunta et al., 2021; Pfeiffer et al., 2022). Adapters are also proposed for task transfer and parameter-efficient fine-tuning (Houlsby et al., 2019; Pfeiffer et al., 2020; Zaken et al., 2022). In addition, parameter averaging (Matena & Raffel, 2022; McMahan et al., 2017; Izmailov et al., 2018; Wortsman et al., 2022; Li et al., 2022; Gururangan et al., 2023), model fusion (Don-Yehiya et al., 2022; Borzunov et al., 2022), continual learning (Jang et al., 2021; Qin et al., 2022; Ke et al., 2022; Qin et al., 2023), and other collaborative approaches (Köpf et al., 2023; Sha, 2023; Luo et al., 2023) have also shed light on the possibility of distributed LM training. However, existing modular LMs mostly operate in the white-box setting, *i.e.* assuming access to the model parameters, token probabilities, and more. Since the most prominent LLMs are only released behind API calls, we propose KNOWLEDGE CARD with the aim of empowering black-box general-purpose LLMs with community-driven and collaborative knowledge.

## 7 CONCLUSION

We propose KNOWLEDGE CARD, a novel framework to empower general-purpose LLMs with modular and collaborative knowledge. We first present knowledge cards, specialized LMs trained on various domains and sources of knowledge, and propose three knowledge selectors to ensure knowledge quality. We then propose *bottom-up* and *top-down* approaches to integrate knowledge cards with general-purpose LLMs to enable multi-domain knowledge synthesis and grounding in external information when necessary. Extensive experiments demonstrate that KNOWLEDGE CARD outperforms vanilla LLMs, retrieval LMs, and generated knowledge prompting approaches across three tasks and six datasets, showcasing its ability to integrate multiple sources of information, efficiently update LLM's knowledge, and more. We envision KNOWLEDGE CARD as a community-driven initiative to empower general-purpose LLMs with modular and collaborative knowledge.

## ACKNOWLEDGEMENTS

We thank the reviewers, the area chair, members of Tsvetshop, and the UW NLP Group for their feedback. This research is supported in part by the Office of the Director of National Intelligence (ODNI), Intelligence Advanced Research Projects Activity (IARPA), via the HIATUS Program contract #2022-22072200004. This material is also funded by the DARPA Grant under Contract No. HR001120C0124. We also gratefully acknowledge support from NSF CAREER Grant No. IIS2142739, NSF Grants No. IIS2125201, IIS2203097, and the Alfred P. Sloan Foundation Fellowship. The views and conclusions contained herein are those of the authors and should not be interpreted as necessarily representing the official policies, either expressed or implied, of ODNI, IARPA, or the U.S. Government. The U.S. Government is authorized to reproduce and distribute reprints for governmental purposes notwithstanding any copyright annotation therein.

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

# A  DISCUSSION

**Modularity at every turn.**   All components in KNOWLEDGE CARD are modular and easily substituted with future state-of-the-art. 1) While Codex is the default LLM in the experiments, KNOWLEDGE CARD also works with TEXT-DAVINCI-003 and GPT-3.5-TURBO (§5) and could be easily adapted to future LLMs. 2) If better models for embedding space similarity, abstractive summarization, and fact-checking are developed, the three knowledge selectors (§2.2) could be seamlessly updated. 3) When new knowledge, information, and domains emerge, more knowledge cards could be trained and uploaded to a model-sharing infrastructure (Wolf et al., 2020) by any member of the machine learning community and adopted to improve general-purpose LLMs.

**User-centric LLM adaptation.**   When general-purpose LLMs are released, everyone uses the same LLM with the same API calls, while real-world users have heterogeneous use cases and expectations that require personalization (Salemi et al., 2023). For example, grade school students might expect LLMs to be absolutely factual about knowledge and information in common textbooks, NLP researchers might expect LLMs to have a basic understanding of current NLP research, cooking amateurs might expect LLMs to understand the basic recipes and cuisines for different occasions, and more. As a result, KNOWLEDGE CARD presents a preliminary approach by letting the user select and activate knowledge cards to empower LLMs with different skill sets and domain expertise.

**Compatible with diversified forms of knowledge.**   By default, KNOWLEDGE CARD employs language models trained on varying domains and corpora as modular knowledge sources. In addition, KNOWLEDGE CARD is also compatible with 1) retrieval systems, where the retrieved text could similarly go through the three knowledge selectors and enrich LLM context, while retrieval corpora are harder to share and use than modular language models; 2) knowledge graphs, when combined with various proposals to construct natural language corpora out of symbolic knowledge bases (Agarwal et al., 2021; Chen et al., 2020; Feng et al., 2023a), which is already included in our prototype; 3) search engines, where content on the web could also be integrated into the black-box LLMs through KNOWLEDGE CARD. Such flexibility and compatibility are possible since KNOWLEDGE CARD conducts knowledge integration through natural language. Compared to retrieval models, using language models as knowledge sources enables flexible information seeking (rather than rigid token exact match), searching over knowledge domains, and employing private knowledge sources.

**Knowledge cards heterogeneity.**   While existing modular LM proposals often require modular sub-models to be of the same size and architecture for parameter averaging and model fusion (Li et al., 2022), the knowledge cards in this work could be fully heterogeneous. 1) Different knowledge cards could have different sizes. While OPT-1.3B is adopted as the default architecture in this work, other sizes of OPT, from 125M all the way up to tens of billions, could all be used to initialize knowledge cards. In addition, 2) knowledge cards could have different model architectures. Since the integration of general-purpose LLMs and modular knowledge cards happens at the natural language level, any language generation models could be adopted as knowledge cards. These two levels of heterogeneity allow for flexibility in knowledge card training: larger and more capable models could be trained on large corpora and extensive knowledge domains by compute-rich individuals, while smaller knowledge cards trained on small and dedicated domains by computationally underprivileged researchers could also help improve black-box LLMs, democratizing LLM research.

**Knowledge cards hierarchy.**   We believe that knowledge cards could reflect the hierarchical nature of knowledge. If KNOWLEDGE CARD is adopted for general question answering, then a general biomedical knowledge card trained on PubMed corpus would suffice. However, if KNOWLEDGE CARD is adopted for more fine-grained use cases, the "biomedical" domain could be further divided into sub-domains and one knowledge card could be trained for each. Similar divisions could be applied to sub-fields in NLP research, political news in different countries, and more.

**Combining bottom-up and top-down.**   One straightforward way to combine the two knowledge integration approaches would be: in each step of top-down, the LLM proposes multiple knowledge cards as candidates, then employs the bottom-up approach with the pool of these knowledge cards for knowledge generation. We leave further explorations to future work.

# B  LIMITATIONS

**Knowledge cards are not perfect knowledge generators.**  While knowledge cards could be of any size or model architecture, we used OPT-1.3B, a relatively small language model to initialize knowledge cards trained on different domains and sources. As a result, not all of the generated knowledge documents are high-quality knowledge statements, occasionally suffering from degeneration, topic deviation, and more. While the three knowledge selectors in part alleviate the impact of low-quality generated knowledge documents, we hypothesize that improving the knowledge generation of autoregressive language models is an important, yet orthogonal, research question for future work. Two potential solutions include 1) increasing the model size of knowledge cards and 2) using specialized training objectives for knowledge cards, while both approaches require additional training and more computational resources. In addition, Appendix A discussed KNOWLEDGE CARD's compatibility with diverse knowledge sources, including retrieval, knowledge graphs, and search engines, while these knowledge repositories have their respective pros and cons. We leave it to future work on integrating multiple types of external knowledge stores to extend KNOWLEDGE CARD.

**The factuality selector is biased towards information-rich domains and existing knowledge.** To ensure the factuality of generated knowledge documents, we employed a retrieval-augmented factuality selector based on both summarization factuality metrics and fact-checking models while enabling flexibility through our proposed *top-k factuality sampling*. However, domains with more Wikipedia entries might be better supported by retrieved documents and might receive higher factuality scores. In addition, new and emerging knowledge might be well supported by existing retrieval corpora and receive low factuality scores. We quantitatively evaluate this bias in Appendix D. Although top-k factuality sampling enables flexibility to some extent, it remains an important problem to design factuality evaluation measures that are generalizable and adaptable to varying and emerging domains.

**Prompting LLMs to seek help through yes/no questions is not perfect.**  Inspired by the findings that LLMs do not need external knowledge for every query (Zhao et al., 2023) and language models (mostly) know what they know (Kadavath et al., 2022), we propose to ask yes/no questions to decide whether to activate knowledge cards and encourage well-calibrated answers through in-context learning. Our analysis (§5) shows that this strategy is effective but far from perfect: LLMs are occasionally over-confident about their knowledge capabilities. We leave it to future work on designing better strategies for LLMs to abstain, acknowledge knowledge limitations, and seek help from external information sources.

# C  ETHICS STATEMENT

We envision KNOWLEDGE CARD as a community-driven and collaborative initiative to improve general-purpose LLMs in knowledge-intensive tasks and contexts. An important risk is the dual use and exploitation from malicious actors. Since modular knowledge cards have the ability to change or update LLM knowledge, malicious actors could advance their agenda by submitting malicious knowledge cards trained on disinformation, hyperpartisan content, propaganda, and more, while framing them as benign knowledge domains and deceive LLM users. We envision two lines of approaches towards this ethical risk: on the *technical* side, research on adversarial manipulation of language models and corresponding defense tactics (Bagdasaryan & Shmatikov, 2022; Perez et al., 2022) could be integrated to alleviate the impact of malicious knowledge cards; on the *social* side, we could rely on and reinforce the existing rules for model sharing on popular infrastructures (Wolf et al., 2020) to prevent such malicious contribution from happening. We encourage the responsible use of KNOWLEDGE CARD, while we also call on users and researchers to be mindful of this dual-use risk.

# D  ANALYSIS (CONT.)

**Knowledge Card Selection**  In the top-down approach, we ask large language models to choose relevant knowledge cards and obtain external knowledge. We illustrate the selection results of the automatic selection strategy on the MMLU dataset separated into the 57 sub-tasks. Figure 8 demonstrates that for most tasks knowledge selection exhibits spike-like patterns on Wikipedia

corpora and encyclopedic knowledge graphs, suggesting that the majority of tasks have a few clearly relevant knowledge cards. In addition, for other tasks (e.g. juris prudence and high school U.S. history), it is not clear which knowledge cards would be most helpful, thus the selection is more spread-out. These results suggest that the selection patterns could also indicate whether a new and more in-topic knowledge card is needed for any given task.

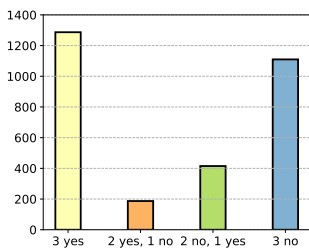

**Yes/No Template Sensitivity** In the top-down approach, we prompt LLMs with "*Do you need more information? (Yes/No)*" to identify if external knowledge is required and use in-context learning to encourage well-calibrated responses. Since language models are sensitive to minor changes in prompts, we devise two more questions: "*Is more information needed here?*" and "*Would you like additional information?*" and report the results on the 2-way misinformation task in Figure 7. It is demonstrated that LLMs give *moderately* consistent responses: 79.9% of cases received unanimous yes or no from the three prompts, while 20.1% examples received mixed results. This suggests that a potential improvement to KNOWLEDGE CARD is to employ multiple yes/no questions to probe knowledge limitations and use an ensemble of answers to improve robustness.

Figure 7: Yes/No questions in *top-down* are mostly consistent across three prompt templates, while there is space for improvement in future work.

**Factuality Scores of Knowledge Cards** We use the MMLU datasets as queries to prompt different knowledge cards, generate knowledge documents, and evaluate their factuality with the factuality selector (§2.2). We illustrate the factuality score distributions of different knowledge cards in Figure 9, which shows that different knowledge cards have varying inherent factuality. We hypothesize that the distribution of factuality scores given by the factuality selector could guide efforts to evaluate the quality of community-contributed knowledge cards.

**Knowledge Card Accumulation** We expect KNOWLEDGE CARD to perform better when relevant knowledge cards are gradually added to the system. To this end, we gradually add five knowledge cards (PubMed, IMDB, Book-Corpus, News, and Wikipedia) to KNOWLEDGE CARD and evaluate performance with the misinformation dataset, 2-way setting, bottom-up approach, and the ChatGPT model. Table 7 demonstrates that the addition of knowledge cards, especially in-domain ones (News in this case), is indeed helpful in improving the base large language model. Table 6: Hyperparameter settings.

| LM Training | | Inference Stage | |
|---|---|---|---|
| **Hyperparameter** | **Value** | **Hyperparameter** | **Value** |
| LEARNING RATE | $2e$-5 | $n_1$ | 3 |
| WEIGHT DECAY | $1e$-5 | $n_2$ | 5 |
| MAX EPOCHS | 10 | $n_3$ | 3 |
| BATCH SIZE | 32 | MAX ITERATION | 1 |
| OPTIMIZER | ADAM | TEMPERATURE | 0.1 |
| ADAM EPSILON | $1e$-6 | DEFAULT LLM | CODEX |
| ADAM BETA | 0.9, 0.98 | | |
| WARMUP RATIO | 0.06 | | |

| Setting | # card | BAcc | MaF |
|---|---|---|---|
| VANILLA | 0 | 80.1 | 70.5 |
| + PUBMED | 1 | 80.7 | 70.6 |
| + IMDB | 2 | 80.6 | 71.2 |
| + BOOKCORPUS | 3 | 82.3 | 72.9 |
| + NEWS | 4 | 85.7 | 73.1 |
| + WIKIPEDIA | 5 | 76.5 | 75.3 |

Table 7: KNOWLEDGE CARD performance on the two-way misinformation dataset when five knowledge cards are gradually added.

**Working Examples** We present the specific prompts, generated knowledge documents, and prompts for the bottom-up approach, and the top-down approach with automatic and explicit selection in Tables 9, 10, and 11 respectively.

# E EXPERIMENT DETAILS

**Algorithm Details** We present an algorithmic summary of the *bottom-up* and *top-down* approach in Algorithm 1 and 2.

**Knowledge Cards Domains** We train a total of 25 knowledge cards from the following corpora and domains: one billion tokens (Chelba et al., 2013), ACL papers (Lo et al., 2020), commonsense knowledge graph ATOMIC (West et al., 2022), book corpus (Zhu et al., 2015), ConceptNet (Speer et al., 2017), biomedical knowledge graph UMLS (Zhang et al., 2021), Gutenberg (Rae et al., 2019),

---

*// open-book setting*
**Question**: Who won the senate race of Oregon in the 2022 U.S. midterm elections?
**Answer**: Ron Wyden

*// two-way setting*
**Question**: Who won the 24th congressional district of Texas in the 2022 U.S. midterm elections?
A. Jan McDowell B. Beth Van Duyne
**Answer**: B

*// four-way setting*
**Question**: Who won the 6th congressional district of Pennsylvania in the 2022 U.S. midterm elections?
A. Christopher Hoeppner B. Doug Mastriano C. Chrissy Houlahan D. Guy Ciarrocchi
**Answer**: C

---

Table 8: Examples of the MidtermQA dataset for the three settings.

IMDB movie reviews (Wang et al., 2023), political knowledge graph KGAP (Feng et al., 2021), legal contracts (Hendrycks et al., 2021), math problems (Saxton et al., 2019), midterm election news (ours), open subtitles [2], political news corpora POLITICS (Liu et al., 2022c), biomedical literature [3], RealNews (Zellers et al., 2019), Reddit (Feng et al., 2023b), Twitter (Feng et al., 2022), Wikidata knowledge graph (Vrandečić & Krötzsch, 2014), Wikipedia dump [4], YAGO (Pellissier Tanon et al., 2020), and Yelp reviews [5]. For knowledge graphs, we follow Feng et al. (2023a) to construct textual corpora and use them as training data.

**Hyperparameters**   We present hyperparameter settings in Table 6.

**Dataset Details**   1) The MMLU dataset (Hendrycks et al., 2020) contains a total of 15,908 four-choice questions further divided into 57 sub-tasks in four domains: humanities, social sciences, STEM, and others. The official dataset also provides a demonstration set, *i.e.* 5-shot examples in each sub-task to enable few-shot in-context learning. We follow the official demonstration set and test set in our experiments. 2) The LUN dataset (Rashkin et al., 2017) is a misinformation detection dataset with two- or four-way classification settings, either with true/false only or fine-grained categories of trusted, hoax, propaganda, or satire. We use the official test set in (Hu et al., 2021) with 2,999 examples throughout the experiments. 3) We curate *MidtermQA*, a QA dataset focusing on the 2022 U.S. midterm elections to evaluate KNOWLEDGE CARD's ability for temporal knowledge update. Specifically, we first collect the results of the 510 races in congressional, senate, or gubernatorial elections in the 2022 midterms. We then construct three datasets: a) open-book, where we ask LLMs to directly answer the name of the election winner for a given race, b) two-way, where we ask LLMs to choose the winner from the two front runners, and c) four-way, where we increase the difficulty by including two other politicians contesting in the same state to create a distraction. We present examples of the MidtermQA dataset in Table 8.

**Computation Resources Details**   We used a GPU cluster with 16 NVIDIA A40 GPUs, 1988G memory, and 104 CPU cores for the experiments. Training knowledge cards took from around 7 hours to 10 days depending on the training corpora size. For the black-box LLMs, we used the OpenAI API to access CODE-DAVINCI-002, TEXT-DAVINCI-003, and GPT-3.5-TURBO in the experiments.

---

[2] https://github.com/sdtblck/Opensubtitles_dataset
[3] https://github.com/thoppe/The-Pile-PubMed
[4] https://github.com/noanabeshima/wikipedia-downloader
[5] https://www.yelp.com/dataset

---

¡in-context examples with the same format¿

...

**Knowledge**:  … San Mateo is located in the northwest of California …   Dianne Feinstein, the senior senator from California, is rumored to retire …   Tom Brady returned to his hometown of San Mateo …

**Question**: Who is the senior senator from Tom Brady's birth place?

**Answer**:

---

Table 9: Prompt example for the bottom-up approach. Different color boxes indicate knowledge documents generated by different specialized LMs.

---

¡in-context examples with the same format¿

...

**Question**: Who is the senior senator from Tom Brady's birth place?

Do you need more information? (Yes or No)

*Yes*

What kind of information do you need?

*The state Tom Brady is from.*

**Knowledge**:  Tom Brady returned to his hometown of San Mateo, CA …

Do you need more information? (Yes or No)

*No*

**Answer**:

---

Table 10: Prompt example for the top-down approach with automatic selection. Italic texts indicate that this field is generated by black-box LLMs.

---

¡in-context examples with the same format¿

...

**Question**: Who is the senior senator from Tom Brady's birth place?

Do you need more information? (Yes or No)

*Yes*

Choose an information source from the following: sports, biomedical literature, NLP papers, book corpus.

*sports*

**Knowledge**:  Tom Brady returned to his hometown of San Mateo, CA …

Do you need more information? (Yes or No)

*No*

**Answer**:

---

Table 11: Prompt example for the top-down approach with explicit selection. Italic texts indicate that this field is generated by black-box LLMs.

---

**Algorithm 1:** Bottom-Up Approach

---

**Data:** question $q$; in-context examples prompt $s_{icl}$; knowledge cards $\mathcal{C} = \{\text{spec}_1, \ldots, \text{spec}_n\}$;
relevance, pruning, and factuality selector $\phi_{rel}, \phi_{prune}, \phi_{fact}$
**Result:** answer string $s_{ans}$
PROMPT = $s_{icl}$
KNOWLEDGE_LIST = []
**for** spec $\in \mathcal{C}$ **do**
  | KNOWLEDGE_LIST.append(spec($q$, $n_1$))
**end**
KNOWLEDGE_LIST = $\phi_{rel}(q$, KNOWLEDGE_LIST, $n_2$)
KNOWLEDGE_LIST = $\phi_{prune}$(KNOWLEDGE_LIST)
KNOWLEDGE_LIST = $\phi_{fact}$(KNOWLEDGE_LIST, $n_3$)
PROMPT += "*Knowledge:* "
**for** $s \in$ KNOWLEDGE_LIST **do**
  | PROMPT += $s$
**end**
PROMPT += "*Question:* " + $q$ + "*Answer:*"
$s_{ans}$ = LLM(PROMPT)
return $s_{ans}$

---

**Algorithm 2:** Top-Down Approach

---

**Data:** question $q$; in-context examples prompt $s_{icl}$; knowledge cards $\mathcal{C} = \{\text{spec}_1, \ldots, \text{spec}_n\}$;
knowledge card names $\mathcal{S} = \{s_1, \ldots, s_n\}$; max trial $k$; relevance and factuality selector
$\phi_{rel}$; $\phi_{fact}$; binary flags AUTO and EXP
**Result:** answer string $s_{ans}$
PROMPT = $s_{icl}$
PROMPT += "*Question:* " + $q$
$i = 0$
**while** $i \leq k$ **do**
  | PROMPT += "*Do you need more information? (Yes or No)*"
  | RESPONSE = LLM(PROMPT)
  | **if** RESPONSE == *"Yes"* **then**
    | **if** AUTO **then**
      | PROMPT += "*What kind of information do you need?*"
      | RESPONSE = LLM(PROMPT)
      | spec = $\phi_{rel}$(RESPONSE, $\{s_1, \ldots, s_n\}$, TOP-K = 1)
    | **end**
    | **if** EXP **then**
      | PROMPT += "*Choose an information source from the following:* "
      | **for** $s \in \mathcal{S}$ **do**
        | prompt += $s$
      | **end**
      | spec = LLM(PROMPT)
    | **end**
    | KNOWLEDGE = $\phi_{fact}$(spec($q$, $n_1$), 1)
    | PROMPT += "*Knowledge:* " + KNOWLEDGE
  | **end**
  | **if** RESPONSE == *"No"* **then**
    | break
  | **end**
**end**
PROMPT += "*Answer:*"
$s_{ans}$ = LLM(PROMPT)
return $s_{ans}$

---

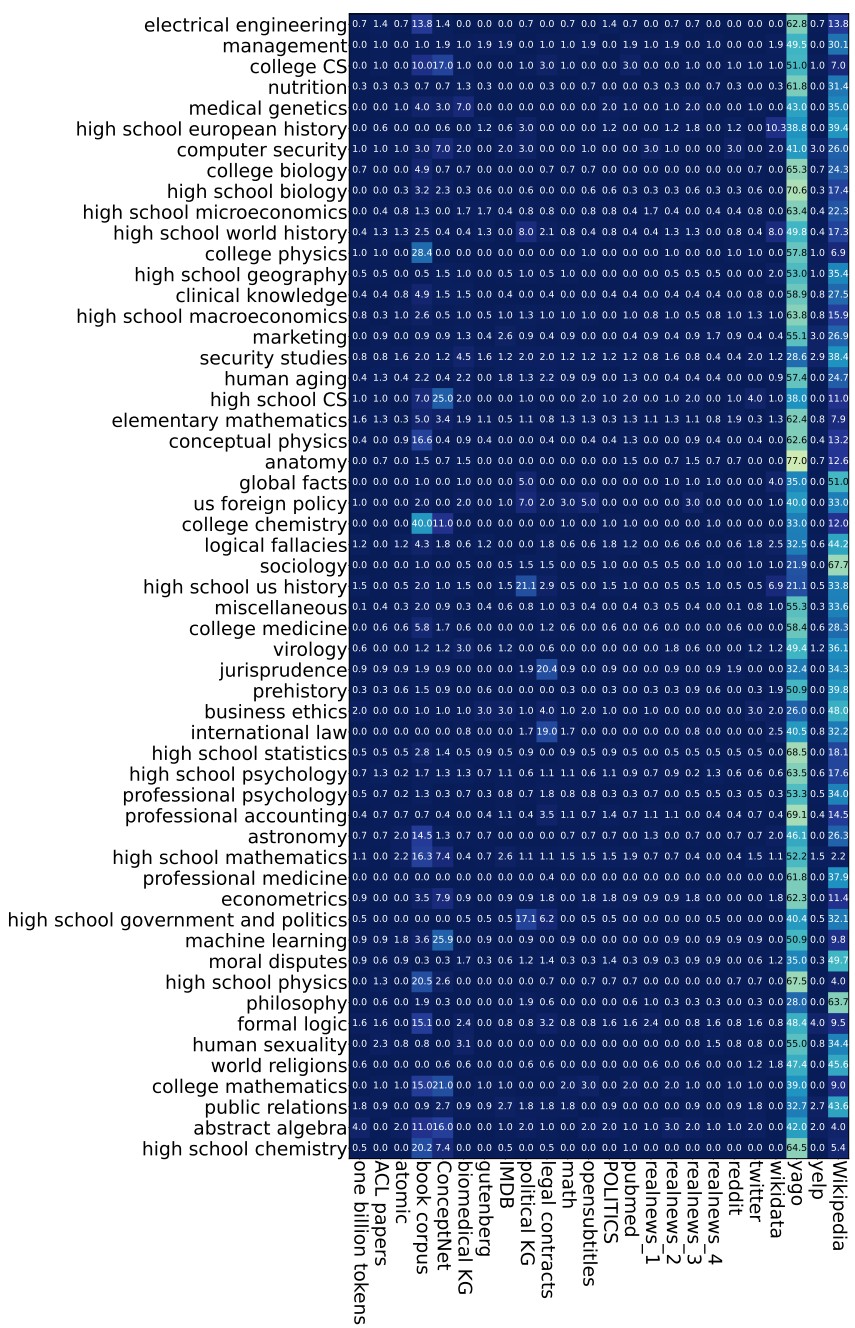

Figure 8: Statistics of knowledge card selection in the top-down approach with automatic selection across 57 sub-tasks in the MMLU benchmark. Encyclopedic knowledge graph YAGO and Wikipedia are generally the most adopted knowledge cards.

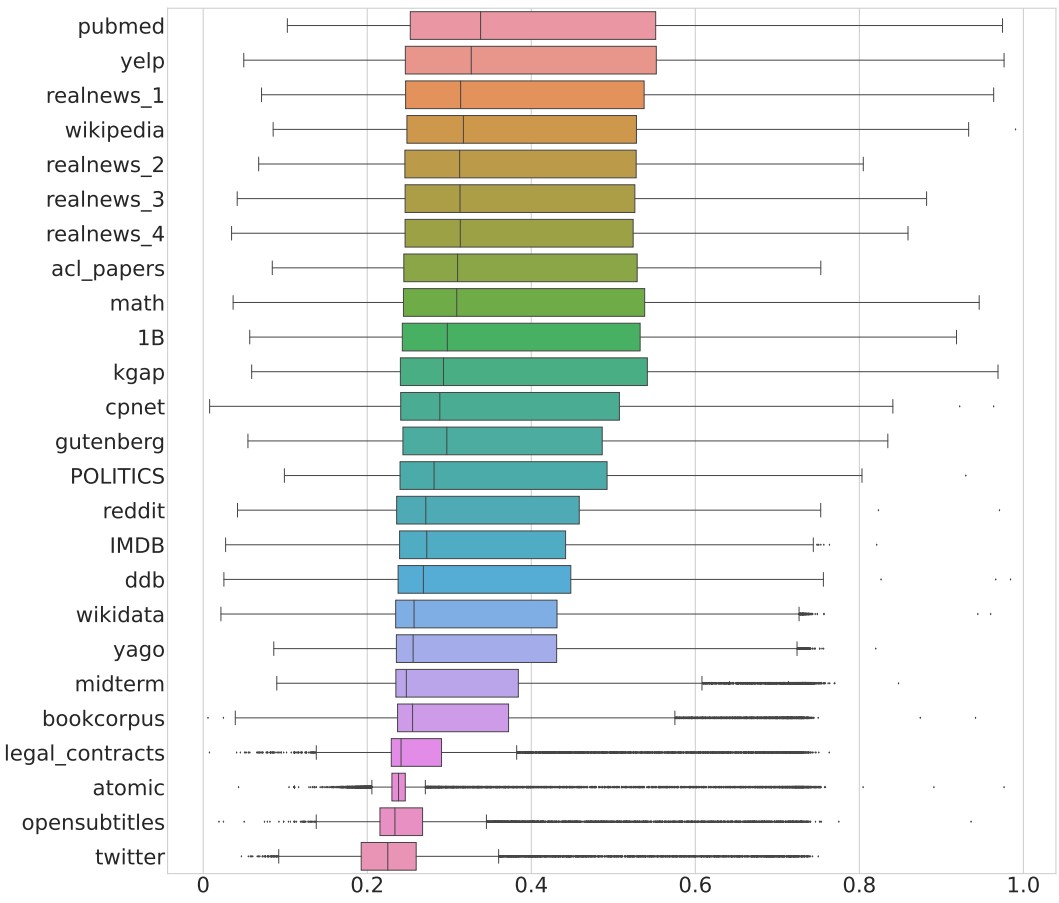

Figure 9: Factuality score distributions of the 25 knowledge cards when prompted with questions in the MMLU benchmark. Different knowledge cards *do* have varying factuality score distributions.

