# OpenReview forum: "Knowledge Card: Filling LLMs' Knowledge Gaps with Plug-in Specialized Language Models"
_ICLR.cc/2024/Conference — ICLR 2024 oral_

### Official Review · Reviewer_HvjK · 2023-10-27

**Soundness:** 3 good
**Presentation:** 4 excellent
**Contribution:** 3 good
**Rating:** 8
**Confidence:** 3

**Summary:**

The paper introduces "Knowledge Card", a modular framework designed to augment large language models (LLMs) with up-to-date, factual, and relevant knowledge. Knowledge cards are trained on specific domains and sources. These knowledge cards act as parametric repositories and are selected during inference to provide background knowledge to the base LLM. The paper claims state-of-the-art performance on six benchmark datasets, demonstrating the effectiveness of the Knowledge Card framework in dynamically synthesizing and updating knowledge across various domains.

**Strengths:**

1. As an alternative for retrieval based method, knwoledge card allows for the dynamic synthesis and updating of knowledge from various domains, which is a significant advancement over static general-purpose LLMs.
2. The method demonstrates state-of-the-art performance on six benchmark datasets. The results indicate that it is beneficial for numerous knowledge-intensive tasks, especially in situations that require the latest and accurate information.
3. Designing the relevance selector, pruning selector, and factuality selector to evaluate dimensions of relevance, brevity, and factuality encompasses a broader and more extensive range than considered in previous methods.

**Weaknesses:**

The benchmarks and datasets tested in the paper primarily focus on natural language understanding tasks, lacking more results on generative tasks.

**Questions:**

The bottom-up approach and top-down approach mentioned in the text each have their own advantages and disadvantages. Is it possible to combine the two?

---

> ### Author Response · Authors · 2023-11-14
> **Author Response**
>
> We would like to thank the reviewer for their thoughtful comments and feedback.
>
> > The benchmarks and datasets tested in the paper primarily focus on natural language understanding tasks, lacking more results on generative tasks.
>
> The three tasks and six datasets in this work focus on classification and open-domain QA tasks (while open-domain QA might be considered as a generation task). While we did not identify good resources to test out LLM knowledge abilities in a long-form generation setting that’s fitting in this work, we believe that Knowledge Card would be better at knowledge generation tasks as well thanks to the modular knowledge cards and the two integration approaches. Our goal is to demonstrate the modular Knowledge Card approach works for knowledge-intensive understanding tasks, while future work can explore expanding it to generative tasks.
>
> > The bottom-up approach and top-down approach mentioned in the text each have their own advantages and disadvantages. Is it possible to combine the two?
>
> One straightforward way to combine them would be: in each step of top-down, the LLM proposes multiple knowledge cards as candidates, then employ the bottom-up approach with the pool of these knowledge cards for knowledge generation. We conducted a quick exploration with this using the gpt-3.5-turbo LLM, finding out that “having LLMs propose multiple knowledge card candidates” would need more than just zero-shot prompting. Thank you for proposing this interesting idea. We would love to explore this in our future work.

---

### Official Review · Reviewer_fqxe · 2023-10-31

**Soundness:** 3 good
**Presentation:** 3 good
**Contribution:** 3 good
**Rating:** 8
**Confidence:** 3

**Summary:**

The authors propose a modular framework augmented with a domain-specific knowledge module called Knowledge Card. They introduce two scenarios (top-down and bottom-up) for knowledge integration. They demonstrate consistent performance improvement across multiple datasets compared to existing retrieval-augmented language models and generated knowledge prompting approaches. They also provide an analysis of each proposed module.

**Strengths:**

- The authors demonstrate improvement across various benchmarks such as general-purpose knowledge QA, misinformation detection, and midterm QA compared to existing retrieval-augmented language models and generated knowledge prompting approaches.
- Through an ablation study, the authors demonstrate the effectiveness of each module and conduct an analysis for each module.

**Weaknesses:**

- Language model based modules appear to entail potential risks. For example, knowledge cards based on a language model necessitate a retrieval-based factuality selector, and inaccuracies can arise in LLM-based yes/no decisions during the top-down knowledge integration process.
- One of the major differences between this work and existing work based on retrieval or generation is the utilization of knowledge cards from multiple domains. Therefore, it would be beneficial to demonstrate performance trends based on the gradual accumulation of knowledge cards or the level of granularity of knowledge cards.

**Questions:**

In Table 1, 2, and 3, there are different performance trends among the three Knowledge Card variations. Do the authors speculate about the potential reasons behind these results?

---

> ### Author Response · Authors · 2023-11-14
> **Author Response**
>
> We would like to thank the reviewer for their thoughtful comments and feedback.
>
> > Language model based modules appear to entail potential risks. For example, knowledge cards based on a language model necessitate a retrieval-based factuality selector, and inaccuracies can arise in LLM-based yes/no decisions during the top-down knowledge integration process.
>
> We agree that each step in Knowledge Card is not, and will never be, 100% accurate. However, we argue that the modular components in Knowledge Card could be seamlessly integrated with the future state-of-the-art and continuously improve. Thanks to the modularity of Knowledge Card, these errors and risks could be continuously mitigated with new factuality evaluation models, strategies for LLMs to abstain and seek information, and more.
>
> > One of the major differences between this work and existing work based on retrieval or generation is the utilization of knowledge cards from multiple domains. Therefore, it would be beneficial to demonstrate performance trends based on the gradual accumulation of knowledge cards or the level of granularity of knowledge cards.
>
> We conduct new experiments to test out knowledge card accumulation with the misinformation dataset, 2-way setting, bottom-up approach, and the ChatGPT model. Starting from 0 knowledge cards (vanilla LLM), we add one knowledge card at a time and reevaluate performance.
>
> |        | vanilla | + PubMed | + IMDB | + BookCorpus | + News | + Wikipedia |
> |:------:|:-------:|:--------:|:------:|:------------:|:------:|:-----------:|
> | # card |    0    |     1    |    2   |       3      |    4   |      5      |
> |  BAcc  |   80.1  |   80.7   |  80.6  |     82.3     |  85.7  |     86.5    |
> |   MaF  |   70.5  |   70.6   |  71.2  |     72.9     |  73.1  |     75.3    |
>
> It is demonstrated that the addition of knowledge cards, especially in-domain ones (News in this case), is helpful in improving the base large language model.
>
> > In Table 1, 2, and 3, there are different performance trends among the three Knowledge Card variations. Do the authors speculate about the potential reasons behind these results?
>
> We argue that the bottom-up and top-down settings of Knowledge Card have their respective pros and cons. While bottom-up is especially good at multi-domain knowledge synthesis and better works with documents spanning multiple knowledge domains, top-down is better at iterative and selective knowledge solicitation. Specifically:
>
> Task 1, MMLU: “top-down generally outperforms bottom-up likely because MMLU contains math-related questions that do not necessitate external knowledge. This observation suggests that top-down
> approaches are better at tasks where external knowledge is not always necessary.”
>
> Task 2, misinformation: “bottom-up outperforms both variants of top-down, thanks to its methodology to jointly activate knowledge cards from various domains and enable multi-domain knowledge synthesis.”
>
> Task 3, MidtermQA: when there is a specific in-domain knowledge card for a given task, the top-down approach is better as it could accurately pinpoint which knowledge card is most needed.
>
> We have included some of the explanations in the current paper and will include these empirical evidence of bottom-up and top-down’s pros and cons in Section 4 of the final version.

---

> > ### Comment · Reviewer_fqxe · 2023-11-20
> >
> > Thank you for the additional experimental results and explanation. I revised the score because I believe the response resolved my curiosity.

---

> > > ### Author Response · Authors · 2023-11-22
> > >
> > > Thank you for your thoughtful comments and suggestions. We are grateful for the opportunity to improve our manuscript based on your feedback :)

---

### Official Review · Reviewer_PFBW · 2023-11-01

**Soundness:** 3 good
**Presentation:** 3 good
**Contribution:** 3 good
**Rating:** 8
**Confidence:** 2

**Summary:**

This work proposes knowledge cards which are essentially language models finetuned on specific domains. These knowledge cards can be probed to generate or recite information the domain-specific LMs have memorized (for a given question) without having to explicitly store encoded representations for each document separately. The authors also propose three knowledge selectors which heuristically govern how to aggregate knowledge from different knowledge cards to generate a final answer. The three knowledge selectors include: 1.) relevance selector chooses relevant (generated) documents given a query 2.) pruning selector to summarize generated documents to fir in the given context length and 3.) factuality selector is used to filter out hallucinating documents based on some entailment score. The authors test their system on several tasks including MMLU, MidtermQA and LUN for hallucination detection.

**Strengths:**

1.) The idea to train individual language models seems novel. Modular knowldge organization can be helpful for making progress on continual learning.

**Weaknesses:**

1.) The paper is missing many details and is hard to follow at times, especially in Section 2.1 and 2.3. (specific issues in questions section)

2.) Existing models that employ similar ideas of modular knowledge organization https://arxiv.org/pdf/2108.05036.pdf, https://arxiv.org/pdf/2203.06311.pdf and datasets that test temporal aspects https://arxiv.org/pdf/2110.03215.pdf are not compared.

**Questions:**

1.) It is unclear how information will be stored in a knowledge card for unseen questions at test time

2.) It appears that an entailment classifier is being used to obtain factuality score, can you elaborate how it is trained?

3.) Can you elaborate more on how the MidtermQA dataset was curated?

4.) In bottom up approach (Figure 1) how would a passage about "San Mateo's senior senator" be looked up on prompting with just the query "Who is the senior senator of Tom Brady’s birth place?". The term "senior senator" will not be sufficient as that would yield a very large number of documents and "San Mateo" term will only be obtained after first step in multi-hop retrieval.

5.) Unlike bottom-up, top-down approach seems iterative where the classifier "Do you need more information?" is used to stop iteration. What does the prompt for this look like? How does it perform on a held-out set.

6.) How is it ensured that all the knowledge cards are being useful? It has been shown that wikipedia can often answer simple questions from other domains, which MMLU dataset often tests.

Typos:
handful of knowledge cardds ->  handful of knowledge cards

---

> ### Author Response · Authors · 2023-11-14
> **Author Response (1/2)**
>
> We would like to thank the reviewer for their thoughtful comments and feedback.
>
> > Existing models that employ similar ideas of modular knowledge organization https://arxiv.org/pdf/2108.05036.pdf, https://arxiv.org/pdf/2203.06311.pdf and datasets that test temporal aspects https://arxiv.org/pdf/2110.03215.pdf are not compared.
>
> We would like to thank the reviewer for valuable pointers to related literature.
>
> Since Knowledge Card “specifically focuses on augmenting **black-box** LLMs to enrich their knowledge capabilities” (Section 1, page 2), while the suggested Demix and ELLE, along with other modular works such as BTM [1] and ColD Fusion [2], operates with **white-box** access to the language model for pretraining and fine-tuning or requiring token probabilities, we argue that they are not comparable. By focusing specifically on the black-box setting, Knowledge Card is compatible with the state-of-the-art proprietary models behind API calls while these approaches are not.
>
> The suggested dataset CKL features a collection of datasets (cc-news, LAMA, invariant-LAMA, etc.) that are curated before 2022, while the knowledge cutoff of our base LLMs (Codex, GPT-3.5, ChatGPT) is after that. We did look at other temporal datasets such as TempLAMA [3] and RealTimeQA [4]. While none of them is especially suitable for creating a temporal misalignment in our work, we propose the MidtermQA dataset, focusing on events that happened in late 2022 and early 2023 to investigate Knowledge Card’s ability to incorporate new and emerging events through a domain-specific knowledge card.
>
> We agree that these suggested works and works on LLM continual learning [5-8] are indeed relevant and we will add them to the related works to cite and discuss these works, better positioning Knowledge Card in their context.
>
> [1] Li, Margaret, et al. "Branch-train-merge: Embarrassingly parallel training of expert language models." arxiv 2022.
>
> [2] Don-Yehiya, Shachar, et al. "ColD Fusion: Collaborative Descent for Distributed Multitask Finetuning." ACL 2023.
>
> [3] Dhingra, Bhuwan, et al. "Time-aware language models as temporal knowledge bases." TACL 2022.
>
> [4] Kasai, Jungo, et al. "RealTime QA: What's the Answer Right Now?." arxiv 2022.
>
> [5] Qin, Yujia, et al. "ELLE: Efficient Lifelong Pre-training for Emerging Data." ACL 2022, Findings.
>
> [6] Jang, Joel, et al. "Towards Continual Knowledge Learning of Language Models." ICLR 2022.
>
> [7] Qin, Yujia, et al. "Recyclable Tuning for Continual Pre-training." arxiv 2023.
>
> [8] Ke, Zixuan, et al. "Continual Pre-training of Language Models." ICLR 2023.
>
> > It is unclear how information will be stored in a knowledge card for unseen questions at test time.
>
> For unseen questions from new and emerging knowledge domains, we incorporate a newly trained knowledge card into the framework to expand its access to new knowledge. For example, the MidtermQA dataset focuses on the news event in late 2022 while LLMs’ knowledge cutoff is earlier than that. We demonstrate that an additional knowledge card trained on midterm election news coverage could significantly improve performance and expand LLM knowledge. (Table 7, Section 4)
>
> > It appears that an entailment classifier is being used to obtain factuality score, can you elaborate how it is trained and if it is
>
> Two entailment classifiers are adopted to obtain the factuality scores.
>
> For **summarization factuality**, we use the state-of-the-art FactKB [1] to evaluate whether the condensed summary accurately reflects the knowledge document generated by knowledge cards. We directly use the publicly available checkpoint of FactKB, which was pretrained on knowledge base data and fine-tuned on summarization factuality datasets.
>
> For **retrieval-augmented factuality**, we use the VitaminC [2] entailment classifier to evaluate whether the generated knowledge is supported by retrieved evidence from Wikipedia. We directly use the publicly available checkpoint of VitaminC, which was trained with contrastive evidence and a spectrum of fact-checking tasks with varying granularity to enhance robustness.
>
> We refer the reviewers to [1-2] for full training details. The review text seems to be cut off, we would be happy to answer the latter half of the question.
>
> [1] Feng, Shangbin, et al. "Factkb: Generalizable factuality evaluation using language models enhanced with factual knowledge." EMNLP 2023.
>
> [2] Schuster, Tal, et al. "Get Your Vitamin C! Robust Fact Verification with Contrastive Evidence." NAACL 2021.

---

> > ### Author Response · Authors · 2023-11-14
> > **Author Response (2/2)**
> >
> > > Can you elaborate more on how the MidtermQA dataset was curated?
> >
> > We have included the full description of the curation of the MidtermQA dataset in Appendix E, *Dataset Details*. A brief overview:
> >
> > - We first collect the metadata, contestants, and outcomes of the 510 races in the 2022 US Midterm Election (Senate, House of Representatives, and Gubernatorial elections) with the help of Wikipedia.
> >
> > - We then construct three settings with these races: *open-book*, where LLMs directly answer with the name of the winner; *two-way*, where LLMs choose the winner from the two frontrunners; *four-way*, where LLMs two more candidates contesting in the same state, but different races, are added to create distractions.
> >
> > - This results in a total of 1530 questions for the MidtermQA dataset. The exact prompt and dataset format is presented in Table 7.
> >
> > > In bottom up approach (figure 1) how would a passage about "San Mateo's senior senator" be looked up on prompting with just the query "Who is the senior senator of Tom Brady’s birth place?". The term "senior senator" will not be sufficient as that would yield a very large number of documents and "San Mateo" term will only be obtained after first step in multi-hop retrieval.
> >
> > Indeed, the bottom-up approach is not specifically designed for multi-hop knowledge reasoning, as it only activates all knowledge cards just once. That’s why we additionally propose the top-down approach, where the knowledge fetching process is multi-hop and iterative.
> >
> > Nevertheless, the bottom-up approach is not useless. It uniquely focuses on **knowledge synthesis**, combining diversified knowledge cards trained on varying domains, refining and condensing generated knowledge into a short passage for LLM QA. The bottom-up approach achieves the best performance on misinformation analysis, showing its capacity to handle multi-faceted documents such as news articles that span diverse knowledge domains.
> >
> > To sum up, bottom-up could be adopted in contexts that span multiple knowledge domains, while top-down is recommended for scenarios with more reasoning requirements than just fact retrieval. Future work could explore automatically selecting the right strategy given the downstream task.
> >
> > > Unlike bottom-up, top-down approach seems iterative where the classifier "Do you need more information?" is used to stop iteration. What does the prompt for this look like? How does it perform on a held-out set.
> >
> > We refer the reviewer to Table 9, Table 10, and Algorithm 2 in the appendix for the specific prompt. To briefly summarize, we use the prompt “*Do you need more information? (Yes or No)*”. In case of a “*yes*” answer, we additionally use either “*What kind of information do you need?*” or “*Choose an information source from the following: <knowledge card domains>*” to selectively activate knowledge cards for the automatic and explicit selection settings respectively.
> >
> > We investigate its effectiveness on the held-out test set in Figure 6. Depending on the difficulty of the question, this mechanism works from 63.02% to 88.23% of the time. While this approach achieves great potential, it is certainly not perfect: we thus call for more research on how to determine if an LLM should request more information for QA and argue that “new approaches to abstain could be easily integrated into Knowledge Card”.
> >
> > > How is it ensured that all the knowledge cards are being useful? It has been shown that wikipedia can often answer simple questions from other domains, which MMLU dataset often tests.
> >
> > We investigate this by how frequently is each card selected for the 57 subtasks in MMLU in Figure 8. While YAGO (general-purpose knowledge graph) and Wikipedia are indeed most frequently activated for knowledge generation, in more domain-specific subtasks such as college chemistry and juris prudence, two other knowledge cards trained on book corpus and legal contracts are also selected to fill in the knowledge gaps of YAGO and Wikipedia. We envision that LLM applications in more domain-specific contexts (biomedical QA, LLM personalization, etc.) would even benefit more from the Knowledge Card framework.
> >
> > > Typos: handful of knowledge cardds -> handful of knowledge cards
> >
> > Thank you for pointing this out: we will fix this typo.

---

> ### Comment · Reviewer_PFBW · 2023-11-22
>
> Thank you for answering my questions. I have raised the scores accordingly.

---

> > ### Author Response · Authors · 2023-11-22
> >
> > Thank you! We are grateful for your comments and feedback :)

---

### Author Response · Authors · 2023-11-20
**Revised Paper Posted**

Dear reviewers,

We are thankful for your constructive comments and feedback: we have incorporated all your suggested edits and posted an updated version. The updates include but are not limited to citing and discussing suggested references, presenting newly added experiments and results, clarifying methodology and experiment design, providing more analysis and discussion on results and future work, and fixing typos. We would appreciate it if you might have any further feedback.

Thank you,
authors

---

### Meta-Review · Area_Chair_JZsY · 2023-12-12

**Metareview:**

The paper under review introduces Knowledge Card, a framework for enhancing large language models (LLMs) with up-to-date factual knowledge from specialized language models called knowledge cards. This framework seeks to address the static nature of LLMs and the challenges associated with retraining them. The paper proposes using these knowledge cards during inference to synthesize and update knowledge dynamically. The authors also present three selectors for relevance, pruning, and factuality to control the quality of the information integrated from the cards.

Reviewers have received the paper positively, acknowledging its strengths in advancing the dynamic synthesis of knowledge for LLMs and improving performance across knowledge-intensive tasks. The paper's clear presentation, the novelty of the approach, and the state-of-the-art performance on benchmark datasets have been highlighted as significant strengths. However, reviewers raised concerns about missing details, which were clarified in the authors' response, and the necessity of comparing them with relevant modular knowledge organization works. Reviewers encouraged the exploration of the paper's potential in generative tasks and suggested a better understanding of the performance trends related to the number of knowledge cards used. The authors have provided a thorough response addressing all the concerns raised by the reviewers.

Overall, the reviewers have converged to an agreement that, after considering the revisions and clarifications, the paper should be accepted.

**Justification For Why Not Higher Score:**

N/A

**Justification For Why Not Lower Score:**

Overall, the reviewers have converged to an agreement that, after considering the revisions and clarifications, the paper should be accepted. Reviewer PFBW increased their score post-author response, indicating satisfaction with the provided clarifications and additional experimental results. Reviewer fqxe also adjusted their score, expressing that the response had resolved their initial concerns. Reviewer HvjK maintained a positive view throughout the review process.

---

### Decision · Program_Chairs · 2024-01-16

Accept (oral)